# Delivering Glioblastoma a Kick—DGKα Inhibition as a Promising Therapeutic Strategy for GBM

**DOI:** 10.3390/cancers14051269

**Published:** 2022-03-01

**Authors:** Benjamin Purow

**Affiliations:** Neurology Department, University of Virginia, Charlottesville, VA 22904, USA; bwp5g@virginia.edu

**Keywords:** Diacylglycerol kinase, DGKα, glioblastoma

## Abstract

**Simple Summary:**

Glioblastoma is the most common and aggressive brain cancer, and there is a desperate need for new therapeutic strategies for this incurable cancer. Inhibition of Diacylglycerol kinase α provides novel direct mechanisms against glioblastoma cells, but also offers the potential to simultaneously boost immune cell activities against glioblastoma. This review provides an updated summary of the promising potential of Diacylglycerol kinase α inhibition against glioblastoma, with potential implications for other cancers as well.

**Abstract:**

Diacylglycerol kinase α (DGKα) inhibition may be particularly relevant for the treatment of glioblastoma (GBM), a relatively common brain malignancy incurable with current therapies. Prior reports have shown that DGKα inhibition has multiple direct activities against GBM cells, including suppressing the oncogenic pathways mTOR and HIF-1α. It also inhibits pathways associated with the normally treatment-resistant mesenchymal phenotype, yielding preferential activity against mesenchymal GBM; this suggests possible utility in combining DGKα inhibition with radiation and other therapies for which the mesenchymal phenotype promotes resistance. The potential for DGKα inhibition to block or reverse T cell anergy also suggests the potential of DGKα inhibition to boost immunotherapy against GBM, which is generally considered an immunologically “cold” tumor. A recent report indicates that DGKα deficiency increases responsiveness of macrophages, indicating that DGKα inhibition could also have the potential to boost macrophage and microglia activity against GBM—which could be a particularly promising approach given the heavy infiltration of these cells in GBM. DGKα inhibition may therefore offer a promising multi-pronged attack on GBM, with multiple direct anti-GBM activities and also the ability to boost both adaptive and innate immune responses against GBM. However, both the direct and indirect benefits of DGKα inhibition for GBM will likely require combinations with other therapies to achieve meaningful efficacy. Furthermore, GBM offers other challenges for the application of DGKα inhibitors, including decreased accessibility from the blood-brain barrier (BBB). The ideal DGKα inhibitor for GBM will combine potency, specificity, and BBB penetrability. No existing inhibitor is known to meet all these criteria, but the strong potential of DGKα inhibition against this lethal brain cancer should help drive development and testing of agents to bring this promising strategy to the clinic for patients with GBM.

## 1. Introduction

Diacylglycerol kinase alpha (DGKα) is one of ten DGK family members that convert diacylglycerols (DAGs) to phosphatidic acids (PAs). Both DAG and PA strongly regulate numerous biologic processes. DAG is important for activating Protein Kinase C enzymes. PA is typically found at low levels relative to other phospholipids, in the cell membrane and elsewhere, but despite its low concentrations it regulates a number of signaling pathways. DGKα has been found to regulate signaling pathways integral to cancer, despite being only one member of the large DGK family—suggesting a general lack of redundant function across the family. PA has been found to control activity of mTOR, Akt, and Erk, while DGKα has been associated with NF-κB, HIF-1α, c-met, ALK, and VEGF [1,2,3,4,5,6,7,8,9]. DGKα is increasingly associated with numerous biologic processes, and accordingly its inhibition may be promising for many pathologies and diseases. This is certainly true for cancer, in which DGKα inhibition may not only have direct action against cancer cells but also stimulate the immune system to better attack cancers. However, this is likely to have varying efficacy for different cancers. Based on prior reports, DGKα inhibition may be particularly relevant for the treatment of glioblastoma (GBM), the most common brain malignancy and one which is incurable with current therapies. This review will discuss how the potential benefits of DGKα inhibition may make it a particularly good fit for the clinical challenges of treating GBM as part of combination regimens.

## 2. Clinical Challenges in Treating GBM

GBM is the most common and most lethal primary brain tumor, causing 12,000–14,000 deaths each year in the U.S. alone [10]. Median survival following diagnosis is approximately 15 months with current therapy including maximal surgical resection, radiation, and temozolomide chemotherapy [11]. Recent approaches with targeted agents have unfortunately not shown a significant impact on overall survival [12,13,14]. The challenges inherent in developing more effective GBM treatments have become increasingly clear, and include its relentless invasiveness, its resistance to standard treatments, its genetic complexity and molecular adaptability, a subpopulation of GBM cells similar to normal stem cells, and the blood-brain barrier (BBB). Profiling efforts have shown GBM to be genetically complex, with numerous potential oncogenic drivers even within a single tumor. Each GBM is likely to have lesions in three primary signaling networks: receptor tyrosine kinases/Ras/Akt, the p53 pathway, and Rb/CDK cell cycle pathway. However, there may be more than one lesion and driver within a given pathway even within a single GBM, and this combined with genetic instability can lead to rapid adaptation to targeted agents and combinations that may show initial activity. This likely helps explain the lack of clinical activity of single-agent or combination targeted therapy. Access to GBM is another major barrier to effective therapy; the BBB, a barrier formed of tight junctions and astrocytic foot processes surrounding brain vasculature, restricts the penetration of numerous treatments into GBM. The BBB and relative immunologic privilege of the brain are doubtless factors in GBM ranking among the immunologically cold cancer types [15]. Clinical trials of various immunotherapeutic strategies in patients with GBM have generally shown poor efficacy, despite some signs of inducing some level of immune response against the GBM [16]. The immune system has some access and communication with the central nervous system (CNS) via glymphatic flow and lymphatics in the meninges [17], but in general it is somewhat compromised relative to that elsewhere. GBMs are also heavily infiltrated with blood-borne macrophages and brain-resident microglia that play a tumor-supportive role, as well as myeloid-derived suppressor cells [18,19]. Regionally, there are high levels of immunosuppressive cytokines such as Transforming growth factor-β (TGF-β) [20]. Overall, this has made GBM as resistant to immunotherapies as it is to other treatment approaches.

## 3. Direct Action of DGKα Inhibition against GBM Cells

DGKα and its product PA are known to help maintain activity of several oncogenic pathways that have been linked to GBM malignancy and viability. These include Ras and Raf, Akt and mTOR, VEGF, and HIF-1, as described above. It is therefore unsurprising that DGKα knockdown and inhibition have been shown to have direct activity against GBM cells. While DGKα inhibitors are highly unlikely to block these individual targets and pathways as effectively or potently as specific inhibitors for each, the combined inhibition of multiple targets—even to a lesser degree—may be a promising approach for a genetically heterogenous cancer such as GBM with diverse signaling drivers. This seems particularly salient given the poor track record of highly specific single agents in clinical trials for GBM, as single agents or in combinations. In a prior report, we showed that mTOR and HIF-1α blockade are particularly important mediators of the direct activity of HIF-1α inhibition in GBM cells [21], but the most relevant downstream mediators may vary across different GBMs. DGKα inhibition appears to act on mTOR through more than one mechanism, including reducing its transcription through regulation of cyclic AMP levels [21].

Notably, DGKα inhibition may offer benefits versus single-agent or combined specific and potent inhibitors of targets such as mTOR and HIF-1α. Dedicated inhibitors of the mTORC1 complex are known to drive feedback loops leading to activation of the upstream driver Akt, which may yield even greater malignant behavior; however, DGKα inhibition has been shown to suppress Akt as well as mTOR activity in GBM cells [21]. DGKα also has unique interactions with the PI3 kinase/Akt/mTOR pathway, mediated in part through the Src oncogene [22]. In addition, mTOR inhibition may have immunosuppressive effects that blunt an anti-GBM immune response, while DGKα inhibition should boost an anti-GBM immune response (to be addressed in more detail later in this review and elsewhere in this issue). Notably, in some settings DGKζ may be a stronger regulator of mTOR than DGKα [23].

DGKα inhibition may also provide traction against a challenging and common resistance mechanism and subtype found in GBM and other cancers as well—the mesenchymal subtype. The Cancer Genome Atlas (TCGA) and other profiling efforts have suggested three GBM subtypes [24,25,26]: proneural (PN), mesenchymal (MES), and classical (CL). The MES subtype is more aggressive, has greater vascularity, may have *NF1* lesions, and has been associated with higher Akt, TGF-β, and NF-κB activity [24,25,26]. MES GBM is highly resistant, and no effective therapies for it yet exist. Recent evidence suggests that upon treatment, PN GBM cells acquire MES features as an escape/resistance mechanism in a manner analogous to the epithelial-mesenchymal transition (EMT) process [27,28] and dubbed proneural-mesenchymal transition (PMT). Not only can GBM cells transition from one subtype to another [29], but individual patient GBMs often include cells of more than one subtype. As with EMT in epithelial cancers, the MES GBM subtype is associated with resistance to radiation, chemotherapy, and targeted therapies [27,30,31]. A recent study indicated that transition to MES GBM also confers resistance to the antiangiogenic drug bevacizumab, and that this can be blocked with the lignan-derived compound honokiol, which has anti-MES effects [32]. MES GBM has much higher expression of key immune checkpoint proteins as well [33,34,35]. Furthermore, MES GBM is more highly infiltrated with myeloid immune cells that have a tumor-supporting phenotype [36]. Taken together, these findings suggest that the MES phenotype may also confer resistance to immunotherapies. Thus, despite the subtype heterogeneity and plasticity within a single GBM, it is vital that we identify means to effectively treat MES GBM due to its involvement in broad treatment resistance. DGKα inhibition may represent one promising means to do so. We have reported that DGKα inhibition has superior efficacy against MES versus non-MES GBM in vitro and in vivo; to our knowledge this was the first report to demonstrate preferential anti-MES GBM activity in vivo [37]. Our findings suggested that the anti-MES activity of DGKα inhibition arises via regulating the prenylating enzyme geranylgeranyltransferase-I (GGTase I), which we showed binds the DGKα product PA. DGKα inhibition blocked GGTase I activity, which in turn suppressed downstream GGTase I targets such as Rho and Rac—which are hyperactivated and help drive the MES phenotype [37]. DGKα inhibition may therefore provide a rare opportunity to attack this resistant subtype/phenotype in GBM, with the potential for synergistic benefits combining DGKα inhibitors with the numerous therapies resisted by MES transition. This could have broad implications for treatment of other cancers as well. 

## 4. Indirect Activities of DGKα Inhibition on GBM—Potential Antiangiogenic Activity

GBM is a highly vascularized tumor, and the anti-VEGF agent bevacizumab remains in wide use for recurrent GBM. Antiangiogenic agents such as bevacizumab have yet to show an overall survival benefit in trials for all patients with GBM. However, they improve quality-of-life and reduce edema, and one retrospective analysis did show an overall survival benefit of upfront bevacizumab in patients with proneural GBM lacking IDH1/2 mutation [38]. Antiangiogenic activity may thus provide some benefits for patients with GBM, and there is reason to believe that DGKα inhibition may help suppress angiogenesis. DGKα has been shown to regulate both VEGF and HIF-1 signaling [3,8]—providing mechanisms to regulate angiogenesis—and DGKα targeting of the mesenchymal phenotype may also have an impact given that MES GBM is highly vascularized and linked to high expression of angiogenic drivers [39]. One prior report has shown a marked antiangiogenic effect of DGKα inhibition against GBM tumors in vivo, but this was in a subcutaneous setting and not in an orthotopic model [21]. While all of this is preliminary, the effects of DGKα inhibition on angiogenesis are intriguing and merit further investigation in GBM and other cancer models.

## 5. Indirect Activities of DGKα Inhibition on GBM—Potential Immunotherapeutic Activity

While the effects of DGKα inhibition on the immune system will be covered in more detail elsewhere in this issue, it is important to review their potential specifically for GBM therapy. As described above, GBM is considered an immunologically “cold” tumor, with a strongly immunosuppressive local milieu as well as challenges such as the BBB and decreased communication between the immune system and brain/GBM microenvironment. The immunologic functions already ascribed to DGKα suggest that its inhibition may help counteract some of the challenges involved in immunotherapy for GBM. DGKα inhibition has been found to reverse T cell anergy, a prominent mechanism by which T cells become inactivated in tumors [40,41]. Other references have also suggested the potential for DGKα inhibition to increase T cell activity against tumors [4,42,43,44,45], and recently DGKα inhibition has been shown to mitigate T cell exhaustion [46]—which may be a more prominent mechanism than anergy for weakening anti-tumor T cell activity. DGKα inhibition also boosts the activity of chimeric antigen receptor-modified T (CAR-T) cells [43], which to date have shown transient but inadequate activity against human GBM [47,48]. In addition, DGKα inhibition appears to promote not only T cell function but also that of natural killer (NK) cells [49], which can display potent anti-GBM activity [50,51,52,53,54].

A new report from our group indicates that DGKα also acts to regulate macrophages; DGKα knockout/knockdown increases macrophage responsiveness to diverse stimuli [55]. This may indicate that DGKα inhibition could turn macrophages and microglia from a pro-GBM phenotype toward an anti-GBM phenotype, but this remains to be demonstrated. If proven true, this added immune benefit of DGKα inhibition could be particularly relevant for cancers such as GBM that typically include large numbers of macrophages and macrophage analogs, a group that also includes pancreatic cancer. However, given that at baseline in GBM the macrophages and microglia act to support GBM, it is possible that enhanced macrophage responsiveness with DGKα inhibition could even worsen this. It therefore may be important to combine DGKα inhibition with another therapy that helps turn macrophages against GBM, with DGKα inhibition potentially enhancing responsiveness of macrophages to the other agent. Given the new report and its potential implications, further work is needed to determine how DGKα inhibition directly affects macrophage phenotype and activity in the setting of GBM and other cancers.

A number of reports have suggested that the addition of DGKζ knockout to DGKα knockout markedly increases T cell-activating effects. In fact, specific knockout of DGKζ may have stronger T cell-promoting effects than specific DGKα knockout, and this translated to greater anti-tumor activity in one report testing three syngeneic tumor models in mice [56]. However, DGKζ knockout or knockdown has not yet been shown to have direct activity against GBM, and inhibitors are not yet widely available. While perhaps yielding greater effects on T cells, DGKζ knockout may have differing effects on other immune cells; one report demonstrates that it reduces M1 polarization in macrophages [57], and M1 polarization typically aligns with anti-tumor activity. Therefore, DGKζ inhibition may not be as good a fit for GBM therapy as is DGKα inhibition, despite the former’s greater effects on T cells.

It is possible that DGKα inhibition could also have beneficial effects on the expression of certain immunosuppressive checkpoint proteins. Interestingly, the treatment-resistant MES GBM subtype has much higher expression of the immune checkpoints PD-L1 and TIM3 than PN GBM [33,34]. In addition, MES GBMs over-express ICOSLG, which suppresses the immune response against GBM and other cancers [35]. Taken together, these findings suggest that the MES phenotype may also confer resistance to immunotherapies, and suggest that the anti-MES activity of DGKα inhibition could reduce expression of immunosuppressive ligands on GBM cells; this could have broad implications for the use of DGKα inhibitors and other anti-MES agents as adjuncts in cancer immunotherapy. However, this remains speculative and must be tested.

## 6. Likely Need for Combination Regimens with DGKα Inhibition against GBM

It is increasingly being recognized that the heterogeneity and adaptability of GBM will likely necessitate the use of combination regimens to achieve meaningful efficacy. DGKα inhibitors may be particularly well-suited for such combinations, with potential to enhance the activity of numerous therapies with direct anti-cancer activity but also immunotherapies. The broad signaling effects of DGKα inhibition, as well as preferential activity against the treatment-resistant mesenchymal phenotype in GBM and other cancers, could translate to a host of promising anti-GBM combinations with a DGKα inhibitor. We previously reported in vitro synergistic activity combining a DGKα-inhibiting compound with radiation against GBM stem-like cells [37], and our unpublished data also suggest broad in vitro synergistic activity against GBM with combinations of a DGKα inhibitor and several classes of anti-tumor agents. 

Combining DGKα inhibition with immunotherapies may be particularly promising for GBM and other immunologically cold cancers. As described above, DGKα inhibition may help break T cell anergy and potentially exhaustion as well, thus eliminating barriers that impede the efficacy of immunotherapies in GBM. DGKα inhibition has already been shown to boost the anti-cancer activity of CAR-T cells in preclinical studies [58], and other studies described above suggest its potential to promote NK cell- or macrophage-based immunotherapies as well. A recent report indicates anti-cancer synergy with the combination of DGKα inhibition and anti-PD1 checkpoint inhibitor activity, and this regimen may also have promise for GBM [59]. Though initial studies of anti-PD1 antibodies in patients with GBM were disappointing, there have been signs of activity in small studies of neoadjuvant anti-PD1 [60,61]; boosting T cell activity further with the addition of DGKα inhibition to neoadjuvant anti-PD1 may thus warrant testing for GBM. While DGKζ inhibition may have a greater impact on T cells than does DGKα inhibition, as noted above, combining an inhibitor of both with other immunotherapies may yield the most powerful effects [62]. Notably, the Bristol Myers Squibb company has already patented a compound that inhibits both DGKα and DGKζ [63]. It will be important to determine whether such an agent results in more dangerous autoimmune side effects in patients, as this might not be evident in mouse studies.

## 7. Current Status of DGKα Inhibitors in the Setting of GBM

Effectively incorporating DGKα inhibitors into regimens for GBM will of course require the development of adequate inhibitor compounds with favorable pharmacological properties, including sufficient penetration of the BBB. The most widely-used tool compounds in the laboratory in past decades, R59022 and R59949, have modest potency and activity against DGKα and also inhibit other targets more potently [64]. However, we noted that they were nearly identical in structure to ritanserin, an abandoned compound found safe in clinical trials for schizophrenia and insomnia [65,66,67]. We predicted that ritanserin would also function as a DGKα inhibitor, and our studies confirmed this [64]. It has attractive pharmacologic properties; it is orally bioavailable, has a 40-h half-life in humans, and crosses the BBB sufficiently [68,69,70]. Ritanserin binds and inhibits 5-HT2A/B serotonin receptors, helping people be more animated but sleep more—two “side effects” that would be welcome in many patients with GBM and other cancers [71,72,73]. Though safe in human trials, ritanserin was not FDA-approved because it was bypassed by other, more effective anti-schizophrenia medications. However, ritanserin has potential to move DGKα inhibition into the clinic; notably, we obtained Orphan Drug designation from the FDA for the use of ritanserin in treating GBM. Importantly for preclinical studies with these compounds, R59949 has been shown to boost T cell activity in both human and mouse T cells, while a recent report indicates that ritanserin has a similar promoting effect only in human and not mouse T cells [74]. The authors provide the interesting hypothesis that this may arise from ritanserin more strongly inhibiting a serotonin receptor that plays a stimulatory role in mouse but not human T cells.

Newer DGKα-inhibiting compounds with different structural bases may be able to provide better potency and specificity than ritanserin and its analogs, as well as other potential advantages. Efforts are underway to further develop such compounds, likely driven in large part by the appeal of the immunotherapeutic potential of DGKα inhibitors in combination regimens for cancer. The compound CU-3 has been reported as a DGKα inhibitor with a novel structure [75], and Bristol Myers Squibb is also developing DGKα inhibitors with a different structural basis. Avoiding the R59022/R59949/ritanserin backbone will eliminate serotonin receptor inhibition and the accompanying side effects, which can include somnolence. To apply one of these newer compounds fully against GBM, it will require adequate BBB penetration; as yet this has not been reported for any of them.

## 8. Conclusions

DGKα inhibition thus offers the highly attractive potential for multifaceted direct action against GBM with simultaneous immune-boosting activities and potential antiangiogenic activity as well (summarized in Figure 1). That being said, it is highly likely that truly effective therapy for GBM will likely require combinations, and this applies to DGKα inhibition as well despite its multiple mechanisms of action. DGKα inhibition may in fact lend itself to numerous synergistic combinations with other anti-cancer drugs, for GBM and other cancers as well. While we have focused here on GBM, there is every indication that DGKα inhibition will be applicable to other brain tumor types as well. Furthermore, many of the same points made here should also apply to cancers outside the CNS. Fulfilling the potential of DGKα inhibition for GBM and other cancers will likely require the development of new small-molecule inhibitors, but such work is underway and in the meantime there may be utility in a repurposed drug such as ritanserin.

## Figures and Tables

**Figure 1 cancers-14-01269-f001:**
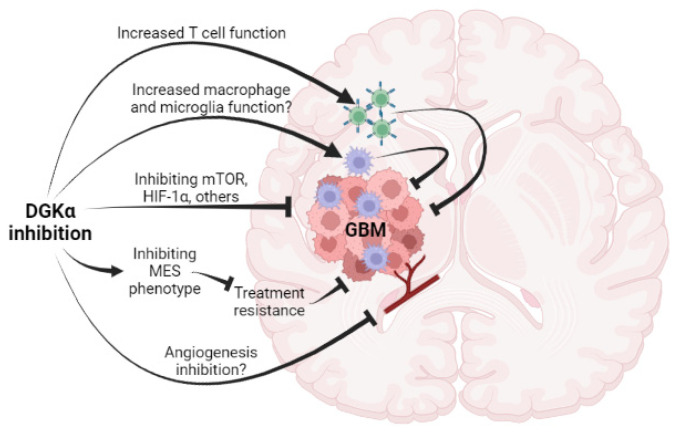
Schematic summary of potential activities of DGKα inhibition against GBM.

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
