# Peer review of "Delivering Glioblastoma a Kick—DGKα Inhibition as a Promising Therapeutic Strategy for GBM"

_cancers, 2022, doi:10.3390/cancers14051269_

Round 1

Reviewer 1 Report

This manuscript is a very well-written and innovative review on the therapeutic benefit of targeting DGKa in glioblastoma. The author provides a comprehensive review of the literature, covering multiple avenues for targeting DGKa. It would be nice to know what the impact of targeting DGKa is on cancer metabolism in the context of GBM and other malignanices. For example, does targeting DGKa (genetically or pharmacologically) induce metabolic changes/adaptations in tumors? Additionally, the manuscript would benefit from a table summarizing known inhibitors of DGKa (clinical/preclinical) in GBM. Very minor typographic errors are highlighted below:

  1. Lines 43 and 46. The greek letters do not show in the pdf version.
  2. Section 2. More references would be helpful to the reader, especially on the immunotherapeutic strategies against GBM.
  3. Line 89. Unless this has been referenced earlier, reference(s) needed for this statement.
  4. Line 277-278. Consider re-writing for clarity.

Author Response

This manuscript is a very well-written and innovative review on the therapeutic benefit of targeting DGKa in glioblastoma. The author provides a comprehensive review of the literature, covering multiple avenues for targeting DGKa. It would be nice to know what the impact of targeting DGKa is on cancer metabolism in the context of GBM and other malignanices. For example, does targeting DGKa (genetically or pharmacologically) induce metabolic changes/adaptations in tumors? Additionally, the manuscript would benefit from a table summarizing known inhibitors of DGKa (clinical/preclinical) in GBM. Very minor typographic errors are highlighted below:

  1. Lines 43 and 46. The greek letters do not show in the pdf version.

Apologies for missing this; it has been corrected in both places.

  1. Section 2. More references would be helpful to the reader, especially on the immunotherapeutic strategies against GBM.

Several references have been added on the immunology of GBM.

  1. Line 89. Unless this has been referenced earlier, reference(s) needed for this statement.

References were added earlier, and mention of this has been added to the text.

  1. Line 277-278. Consider re-writing for clarity.

The sentence has been modified for clarity, thank you.

Reviewer 2 Report

This is a great review highlighting the potential of DGKalpha inhibition against GBM written by an expert in this field.  I strongly feel that the review is well-organized, comprehensive, thus targeting broad audience. In the manuscript, the author not only explain clinical challenge in treating GBM (one of the most immunologically ”cold” cancers with a very limited accessibility due to BBB) in brief, but also discuss the direct and indirect action of DGKalpha inhibition that includes the latest findings: the effects of DGKalpha inhibition on macrophage responsiveness, the effects on immunosuppressive checkpoint proteins, and the current status of inhibitor development, which I believe help the other scientists update their knowledge on the benefit of DGKalpha inhibition on GBM and also attract more interest to pathophysiological roles of DGKalpha.

Minor comments:

Lines 43 and 46 – there are some typo (missing characters) in the manuscript.

Lines 231 – Reference should be specified for the sentence “A recent report indicates anticancer~”.

Author Response

Dear reviewers and editors,

I have revised the review article manuscript as suggested by the reviewers. Please see below for point-by-point replies. Thank you for your time.

Lines 43 and 46 – there are some typo (missing characters) in the manuscript.

This has been corrected, thank you.

Lines 231 – Reference should be specified for the sentence “A recent report indicates anticancer~”.

Apologies for missing this; it has been added.

Reviewer 3 Report

The Review by Purow delineates an interesting hypothesis that stems from recent data, mostly from his group, related to the involvement of Diacylglycerol kinase α (DKGa) in the pathogenesis of glioblastoma (GBM). GBM is the most deadly brain tumor, with a dismal prognosis and almost no curative treatment. Hence, any contribution to unravel any potential pathway that may lead to GBM growth is more than welcome, since it may contribute to pave the way to innovative treatments. 

In this short review, Purow uses a plain and language to drive the reader through the most recent literature on the possible, hopefully curative, effects of DKGa inhibition. DKGa directly affects different oncogenic pathways, particularly relevant in GBM development and may indirectly impact on GBM growth by affecting angiogenesis and immunosuppression. Hence, its inhibition would have an impact on GBM growth.

My minor suggestion is to add a reference at line 105, on the suppression of AKT and mTOR by DKGa inhibition. Also, at lines 237 and 268 the Author states that a pharmaceutical company has patented a compound acting on DGKa: it would be interesting to add a reference for the readers.

Lastly, the Author suggest an interesting hypothesis, on the differential involvement of DKGa in the different subtypes of GBM. What I find lacking is some indication on DKGa itself on the different GBM subtypes: is this information available in published papers? Otherwise, a simple search with just one click on freely available online databases (e.g. TCGA, searched for example with Betastasis, but many more are available) on the expression of genes in the different subtypes of GBM may give some indication to support this hypothesis. A figure on this could be interesting for the reader, if published papers on gene expression or proteins level are not available.

Author Response

Dear reviewers and editors,

I have revised the review article manuscript as suggested by the reviewers. Please see below for point-by-point replies. Thank you for your time.

My minor suggestion is to add a reference at line 105, on the suppression of AKT and mTOR by DKGa inhibition. Also, at lines 237 and 268 the Author states that a pharmaceutical company has patented a compound acting on DGKa: it would be interesting to add a reference for the readers.

The reference was added at line 105; apologies for missing this.

With respect to the patent from BMS, I found it online and have included the link in a reference.

Lastly, the Author suggest an interesting hypothesis, on the differential involvement of DKGa in the different subtypes of GBM. What I find lacking is some indication on DKGa itself on the different GBM subtypes: is this information available in published papers? Otherwise, a simple search with just one click on freely available online databases (e.g. TCGA, searched for example with Betastasis, but many more are available) on the expression of genes in the different subtypes of GBM may give some indication to support this hypothesis. A figure on this could be interesting for the reader, if published papers on gene expression or proteins level are not available.

This is an excellent point. We have checked in the past and there are not significant differences in DGKα expression across GBM subtypes, but our prior report indicates markedly different impact of DGKα inhibition across different subtypes/phenotypes as described (so we left as is).